# Discovery and Optimization of Neutralizing SARS-CoV-2 Antibodies Using ALTHEA Gold Plus Libraries™

**DOI:** 10.3390/ijms24054609

**Published:** 2023-02-27

**Authors:** Omar U. Guzmán-Bringas, Keyla M. Gómez-Castellano, Edith González-González, Juana Salinas-Trujano, Said Vázquez-Leyva, Luis Vallejo-Castillo, Sonia M. Pérez-Tapia, Juan C. Almagro

**Affiliations:** 1Unidad de Desarrollo e Investigación en Bioterapéuticos (UDIBI), Escuela Nacional de Ciencias Biológicas, Instituto Politécnico Nacional, Prolongación de Carpio y Plan de Ayala S/N, Colonia Santo Tomás, Alcaldía Miguel Hidalgo, Mexico City 11340, Mexico; 2Laboratorio Nacional Para Servicios Especializados de Investigación, Desarrollo e Innovación (I + D + I) Para Farmoquímicos y Biotecnológicos, LANSEIDI-FarBiotec-CONACyT, Prolongación de Carpio y Plan de Ayala S/N, Colonia Santo Tomás, Alcaldía Miguel Hidalgo, Mexico City 11340, Mexico; 3Departamento de Inmunología, Escuela Nacional de Ciencias Biológicas, Instituto Politécnico Nacional, Prolongación de Carpio y Plan de Ayala S/N, Colonia Santo Tomás, Alcaldía Miguel Hidalgo, Mexico City 11340, Mexico; 4GlobalBio, Inc., 320 Concord Ave, Cambridge, MA 02138, USA

**Keywords:** COVID-19, phage display, therapeutic antibodies, affinity maturation, semi-synthetic libraries

## Abstract

We recently reported the isolation and characterization of anti-SARS-CoV-2 antibodies from a phage display library built with the VH repertoire of a convalescent COVID-19 patient, paired with four naïve synthetic VL libraries. One of the antibodies, called IgG-A7, neutralized the Wuhan, Delta (B.1.617.2) and Omicron (B.1.1.529) strains in authentic neutralization tests (PRNT). It also protected 100% transgenic mice expressing the human angiotensin-converting enzyme 2 (hACE-2) from SARS-CoV-2 infection. In this study, the four synthetic VL libraries were combined with the semi-synthetic VH repertoire of ALTHEA Gold Libraries™ to generate a set of fully naïve, general-purpose, libraries called ALTHEA Gold Plus Libraries™. Three out of 24 specific clones for the RBD isolated from the libraries, with affinity in the low nanomolar range and sub-optimal in vitro neutralization in PRNT, were affinity optimized via a method called “Rapid Affinity Maturation” (RAM). The final molecules reached sub-nanomolar neutralization potency, slightly superior to IgG-A7, while the developability profile over the parental molecules was improved. These results demonstrate that general-purpose libraries are a valuable source of potent neutralizing antibodies. Importantly, since general-purpose libraries are “ready-to-use”, it could expedite isolation of antibodies for rapidly evolving viruses such as SARS-CoV-2.

## 1. Introduction

The devastating impact of coronavirus disease 2019 (COVID-19) caused by the severe acute respiratory syndrome coronavirus 2 (SARS-CoV-2) on human health and the global economy prompted an unprecedented search for diagnostic, prophylactic and therapeutic options to control this viral infection. Due to the success of antibodies in preventing and treating diverse infectious diseases [1,2], hundreds of academic laboratories as well as small, medium and large biotech companies around the world focused their research efforts on isolating and characterizing anti-SARS-CoV-2 antibodies. Such efforts led to the Emergency Use Authorization (EUA) by the Food and Drug Administration (FDA) and/or the European Agency of Medicines (EMA) of three cocktails of two antibodies each plus three standalone prophylactic and/or therapeutic anti-COVID-19 antibody-based drugs [3]. Nevertheless, as SARS-CoV-2 continues to evolve into new variants of concern (VOCs) and immune scape variants [4], EUA antibodies have lost efficacy, resulting in a continuous quest for new diagnostic, prophylactic and/or therapeutic antibodies to manage COVID-19.

The variable (V) regions serving as substrate for engineering the EUA antibodies were obtained from immune repertoires of COVID-19 convalescent and/or infected patients, mostly via B-cell selection and V region cloning, with one EUA antibody, Regdanvimab, isolated from an immune phage display library [5]. This has been in part due to the suggestion [6] that immune repertoires commonly generate more potent neutralizing antibodies than those obtained from naïve, general-purpose, libraries. However, the latter have advantages over immune repertoires that can be exploited to expedite the isolation of antibodies for rapidly evolving viruses. Among others, libraries built with general-purpose antibody repertoires are “ready-to-use”, avoiding the search for sources of immune repertoires and library construction, thus shortening the discovery phase of V regions to be used as substrate for engineering antibody-based drugs. More importantly, learning from hundreds of therapeutic antibodies that have failed in preclinical and clinical development, the latest generation of general-purpose synthetic or semisynthetic libraries [7] have been designed to maximize the developability profile of the selected V regions including—but not limited to—expression, aggregation, solubility and chemical and long-term stability, which translates into molecules that could speedily and robustly be developed in therapeutic drugs.

We reported in previous works [8,9] the characterization of highly potent antibodies with broad SARS-CoV-2 neutralizing capacity obtained from an immune scFv phage-displayed library. This library was built with the VH repertoire of a convalescent COVID-19 patient infected with the SARS-CoV-2 Delta (B.1.617.2) variant, who was previously vaccinated with a single dose of Convidecia™. Four synthetic VL libraries were used as counterpart of this patient immune VH repertoire. Two of the VL libraries were built with the IGKV4-01 and IGKV3-20 human germline genes, which were designed and tested as part of ALTHEA Gold Libraries™ [10]. The other two VL libraries were designed with the IGKV1-39 and IGKV3-11 human germline genes. These additional VLs increased the structural diversity of the libraries with the potential of generating a more diverse set of antibodies.

After panning the library with the SARS-CoV-2 receptor-binding domain of (RBD) wild-type (WT), also known as the Wuhan variant, we obtained a panel of 34 anti-SARS-CoV-2 scFvs encoded by diverse IGHV germline genes, combined with variants of all the four synthetic VL libraries. Several antibodies blocked the interaction between the RBD WT and the human angiotensin-converting enzyme 2 (hACE-2). One of them, called IgG-A7, neutralized the SARS-CoV-2 Wuhan, Delta and Omicron (B.1.1.529) strains in authentic neutralization tests (PRNT) and protected 100% transgenic mice expressing hACE-2 from SARS-CoV-2 infection at a 0.5 mg/kg dose [9]. These results demonstrated the potential of IgG-A7 for developing broadly neutralizing anti-SARS-CoV-2 prophylactic and/or therapeutic drugs, as well as the value of the immune library as a source of potent neutralizing antibodies.

In parallel to the construction of the immune library, we built a set of naïve general-purpose libraries called ALTHEA Gold Plus Libraries™ by combining the four synthetic VL libraries used in the immune library with a semi-synthetic VH repertoire, which was also designed and validated as part of ALTHEA Gold Libraries™ [10]. This VH repertoire consisted of the well-known human IGHV3-23 germline gene diversified at HCDR1 and HCDR2 positions in contact with proteins and peptide antigens. The HCDR3 diversity was obtained from human HCDR3/JH (H3J) fragments RT-PCR amplified from peripheral blood mononuclear cells (PBMCs) of 200 healthy human donors.

In this study, we report the isolation and characterization of potent neutralizing SARS-CoV-2 antibodies from ALTHEA Gold Plus Libraries™. Since the immune library and ALTHEA Gold Plus libraries™ shared the same synthetic VL libraries, the panning conditions were similar, and the target (RBD WT) used for the selections was the same, the comparison of the antibodies generated from these libraries offered a unique opportunity to assess the performance of naïve, general-purpose libraries, versus an immune library. The lessons learned from this comparison are discussed.

## 2. Results

### 2.1. ALTHEA Gold Plus Libraries™ Construction and Quality Control

The design and implementation ALTHEA Gold Plus Libraries™ has been described in detail at Almagro and Pohl [11]. In brief, the four synthetic VL libraries used in the immune library were combined with the VH repertoire of ALTHEA Gold Libraries™ [10]. The functionality, stability and diversity of the resultant libraries were improved throughout a three-step construction process. In a first step, fully synthetic primary libraries (PLs) containing the four synthetic VL libraries, the universal VH3-23 diversified scaffold and 90 neutral H3J fragments were cloned into the phage display vector as scFvs in a VL-linker-VH configuration. The second step consisted of selecting thermostable scFvs from the PLs, based on the natural capacity of the Protein L of *Peptostreptococcus magnus* to bind the framework 1 of the synthetic VL libraries [12], after a heat shock of 10 min at 55 °C. In the third and final step, the thermostable synthetic antibody fragments, called filtrated libraries (FLs), were PCR amplified and combined with the natural H3J fragments RT-PCR amplified from 200 healthy human donors, thus generating secondary libraries (SLs) called ALTHEA Gold Plus Libraries™.

The filtration process with Protein L differed from the Protein A filtration process reported for ALTHEA Gold Libraries™ [10]. Hence, before cloning ALTHEA Gold Plus Libraries™, the thermostability of the four scFv scaffolds and binding to Protein L were tested (Figure 1). Although the thermostability profile of the scaffolds obtained by Protein L resembled that of Protein A, i.e., 1-39/3-23 scFv was the most stable scaffold, whereas 4-01/3-23 was the least stable one, significant differences depending on using Protein L or A in the unfolding dynamics were observed. For instance, with Protein L 1-39/3-23 seemed to be very stable all along the range of temperatures analyzed. In contrast, with Protein A, although 1-39/3-23 seemed to be more stable than the other scaffolds, it started unfolding at ~60 °C, reaching a Tm of ~72 °C.

Having characterized the thermostability profile of the four scFv scaffolds as well as binding to Protein L, the synthetic scFv fragments containing the PLs were cloned in the phage display vector pADL-23c, generating libraries of ~10^9^ colony transforming units (cfu). Ten individual clones chosen at random from each PL were submitted to Sanger sequencing, showing that all the scFv sequences matched the design and were different at diversified CDR positions, with ~70% of the clones being in-frame sequences.

After the heat shock and rescuing the well-folded scFv variants with Protein L, the number of clones in the FLs was ~5 × 10^9^ cfu. Considering that PLs were ~10^9^ cfu, the number of rescued cfu in the FLs assured a good coverage (~5×) of the PLs diversity. After cloning the natural H3J fragments and generating the SLs (ALTHEA Gold Plus Libraries™), the diversity reached over 10^10^ cfu, with over 90% of in-frame clones and no stop codons.

### 2.2. Selection of Anti-SARS-CoV-2 Neutralizing Antibodies

After three rounds of panning in solid phase with RBD WT as selector, 630 clones were tested for binding to RBD, yielding 125 (19.8%) positive and specific scFvs, with 24 (3.8%) being unique clones, as assessed by Sanger sequencing. A summary of the selection frequency of the unique scFvs, VL scaffolds, HCDR3 lengths and sequences, and ELISA binding to RBD, is provided in Table 1.

Several clones were selected more than once, with one clone, E4R3, outperforming the selections with a frequency of 51%. All four VL scaffolds used to build ALTHEA Gold Plus Libraries™ were seen in the 24 unique clones. The 1-39 scaffold was the most frequent, with 38% of the clones showing variants of this scaffold; it should be noted that the prevalence of the 1-39 scaffold corelated with the performance of this scaffold in the thermostability test (see above). The other three scaffolds were found in 21% (3-11), 13% (4-01) and 13% (3-20) of the clones. Diverse HCDR3 lengths were also found in the unique clones, with a range of lengths of seven to twenty residues. The predominant length was nine residues (21%), followed by eleven and fourteen residues with 13% each.

The top eleven clones in terms of OD signal in the RBD-ELISA were converted to hIgG1 for further characterization. After expression in HEK293 cells and Protein A purification, RBD binding, RBD:hACE-2 blocking and competition of the antibodies with hACE-2 for RBD binding were assessed. Three antibodies (P4A9, P5E1 and P5A10) demonstrated the best binding profile to RBD and blocked the RBD:hACE-2 interaction. However, monomeric content and integrity based on SDS-PAGE of P5E1 and P5A10 were superior to P4A9. Therefore, we focused our further characterization on P5E1 and P5A10.

The monomeric content, molecular weight estimated by SDS-PAGE and thermal stability of P5E1 and P5A10 are shown in Table 2, together with IgG-A7 and CB6 as reference antibodies. The monomeric content of P5E1 and P5A10 was >95%, with bands of the expected molecular weight in SDS-PAGE. The thermostability profile of P5E1 and P5A10 was similar, with two main unfolding transitions; one at 64–69 °C, corresponding to unfolding of the C_H_2 domain [13], and a second transition around 80 °C, which should correspond with the Fab/C_H_3 unfolding, indicating that the Fabs of both antibodies were highly stable.

The functional assessment of P5E1 and P5A10 showed that the former was a tighter binder than the latter, with EC_50_ values in the binding ELISA of 0.97 and 16.87 nM, respectively. The EC_50_ of IgG-A7 and CB6 were 0.025 and 0.027 nM, which were ~40-fold and ~670-fold better than that of P5E1 and P5A10, respectively. In the competition assay, the IC_50_ of P5E1, P5A10, IgG-A7 and CB6 were 2.15, 2.34, 0.19 and 0.49 nM, respectively. In the blocking assay, the IC_50_ of P5E1, IgG-A7 and CB6 were 2.63, 0.25 and 1.29 nM, respectively. Interestingly, P5A10 did not block the RBD:hACE-2 interaction, suggesting that binding to the RBD was not good enough to compete with hACE-2 and/or P5A10 recognized a different epitope than that of P5E1, IgG-A7 and CB6.

On the other hand, the K_D_ values obtained in Biacore were 27.09 nM, 28.78 nM, 0.68 and 12.06 nM for P5E1, P5A10, IgG-A7 and CB6, respectively. The K_D_ reported [14] for CB6 was 2.49 nM. The difference with the value reported in this work could have been in part due to the difference in the conditions used in our K_D_ measurements with respect to those used by Shi et al. [14]. Among other differences, in our study, the antibodies were captured with an anti-IgG antibody, whereas Shi et al. used a Protein A biosensor chip.

PRNT showed that although P5E1 and P5A10 neutralized the virus, it only reached 80% neutralization at a concentration of 100 µg/mL. As previously reported [8] and confirmed in this report, IgG-A7 and CB6 showed 100% neutralization of SARS-CoV-2 WT at 100 µg/mL, with NC_50_ values of 0.56 and 0.56–2.74 nM, respectively. A lower neutralization potency of P5E1 and P5A10 with respect to IgG-A7 and CB6 was consistent with the lower affinity and competition activity of these antibodies when compared to IgG-A7 and CB6. This compelled us to further optimize the affinity of both P5E1 and P5A10 antibodies.

### 2.3. P5E1 and P5A10 VH RAM

To increase the affinity of P5E1 and P5A10, two RAM libraries were built: P5E1_CDRH1/2 and P5A10_CDRH1/2 libraries. These libraries combined the light chains and HCDR3s of P5E1 and P5A10 with the VH synthetic libraries prior to the filtration process (see above). Selection for higher affinity variants was performed via solution panning using decreasing concentrations of biotinylated RBD WT and competition with non-biotinylated RBD WT. Screening for binding to RBD of 45 clones selected from the third round of panning of each RAM library yielded ~50% positive and unique clones for both libraries: 20 clones from the P5E1_CDRH1/2 library and 21 from the P5EA10_CDRH1/2 library.

The scFvs with higher ELISA signals than the parental molecules in the RBD binding assay were converted to hIgG1. Based on the binding profile as hIgG1, one antibody (P5E1-A6) was selected for further development from the P5E1_CDRH1/2 library and two antibodies (P5A10-G2 and P5A10-G4) from the P5EA10_CDRH1/2 library. The three antibodies slightly improved the monomeric content (Table 3). The K_D_ values reached 0.14–0.89 nM. As expected, improvement in affinity translated into better blocking/competition activities, and importantly, higher neutralization potency, with NC_50_ values of 10.45–61.10 nM.

### 2.4. P5E1-A6 VL RAM

To explore whether we could increase the affinity and neutralization potency further, we reshuffled diversity at VL by combining the VH sequence of P5E1-A6 with the synthetic library corresponding to its VL (4-01) prior to the filtration process. After following a selection procedure more stringent than for RAM VH, i.e., lower antigen concentration, two rounds of panning in solution, and overnight incubation plus competition with non-biotinylated RBD, three clones (P5E1-A6-A4, P5E1-A6-E2 and P5E1-A6-E6) were converted to hIgG1 and further characterized. The three P5E1-A6 variants retained the developability profile of the parental molecule and increased the neutralization potency to the sub-nanomolar range (Figure 2), with NC_50_ values of 0.37–0.55 nM, which was two orders of magnitude higher than the neutralization potency of P5E1-A6 (42–45 nM), superior to CB6 (0.56–2.74 nM) and with one of the molecules (P5E1-A6-E6) proving to be slightly better than IgG-A7 (0.56 nM; see Table 2).

### 2.5. Sequence Analysis of P5E1-A6, P5A10-G2 and P5A10-G4

The amino acid replacements that occurred during the stepwise selection and optimization processes reported above are depicted in Figure 3. P5A10-G4 had an insertion of five amino acids “GRRAF” in the HCDR2 with respect to the VH3-23 scaffold. P5E1-A6 had a deletion of one amino acid at HCDR2. P5A10-G2 had three mutations in the HCDR1 and one in HCDR2 with respect to the parental P5A10 but did not have indels. P5A10-G4 and P5E1-A6 indels were not part of the library design [10] and thus, should have been introduced during the synthesis of the PLs or the overlapping PCR assemblage of the RAM libraries. Selection of indels leading to significant improvements in affinity has previously been reported [15]. However, the stretch of five additional amino acids in the HCDR2 of P5A10-G4 has not been seen in the repertoire of human IGHV germline genes [16,17] and thus, may be immunogenic when the antibody is used in human therapy. Therefore, P5A10-G4 was not further characterized.

The deletion of one amino acid at the HCDR2 of P5E1-A6 converted the canonical structure from type 3 in VH3-23 scaffold to type 1 in P5E1-A6 [18,19,20]. Canonical structure type 1 is typical of some human genes in family IGHV3 such as IGHV3-53 and IGHV3-66. Remarkably, antibodies encoded by IGHV3-53/3-66 germline genes and having short HCDR3 loops have frequently been found in anti-RBD antibodies [21]. In fact, we selected from the immune library [8] seven RBD-specific antibodies encoded by the IGHV3-53 germline gene with relatively short HCDR3 (11–12 residues) out of a total 34 (20%). All of them were functionally clustered with IgG-A7, competed with P5E1-A6 and blocked the RBD:hACE-2 interaction, suggesting a similar binding mechanism to the RBD.

Regarding the VL RAM selection, one mutation of glycine (G) to alanine (A) in the LCDR2 of all the three affinity matured antibodies, and one mutation in the LCDR3 of serine (S) to glutamic acid (E) or a double replacement of serine-threonine (ST) to asparagine (N)-S, were responsible for improving the affinity of P5E1-A6-A4 and P5E1-A6-E2, respectively. In P5E1-A6-E6, in addition to the E substitution, a mutation of S to Tyrosine (Y) was seen in the LCDR1. The G to A substitution in LCDR2 might have lowered the flexibility of loop and perhaps impacted its conformation. Introducing a negative change (E) in LCDR3 may have led to a salt bridge between the antibodies and positive charges in the RBD, thus strengthening the interaction between the molecules and, hence, the neutralization potency.

### 2.6. Epitope Mapping and Mechanism of Neutralization of P5E1-A6, P5A10-G2 and IgG-A7

Finally, to understand the region of the RBD bound by P5E1-A6, P5A10-G2 and IgG-A7 and thus infer the mechanism of neutralization, we determined the epitopes recognized by these antibodies using mass spectrometry cross-linking DSS MALDI MS analysis (Figure 4). P5E1-A6 and P5A10-G2 epitopes overlapped with the core RBD region contacted by hACE-2. IgG-A7 mostly had contacts located in the periphery of the RBD interface with hACE-2, suggesting a mechanism of neutralization by sterically blockading the RBD interaction with hACE-2 rather than binding residues in the core of the RBD:hACE-2 interface as P5E1-A6 and P5A10-G2.

The elucidation of the epitopes recognized by P5E1-A6, P5A10-G2 and IgG-A7 also shed light on previously reported functional data [8] showing that: (1) P5E1-A6 competes with IgG-A7 for binding to RBD and (2) IgG-A7 binds the RBDs of the Wuhan, Delta and Omicron variants, whereas P5E1-A6 binds the RBDs of the Wuhan and Delta variants, but not Omicron variant. Actually, P5E1-A6 and P5A10-G2 epitopes on the RBD showed that these antibodies bind several of the residues in the RBD WT that are mutated in the Omicron variant. In contrast, the IgG-A7 epitope mostly mapped onto residues in RBD WT that were not changed in the Omicron variant. Therefore, Omicron mutations seemed to have abrogated binding to P5E1-A6 and P5A10-G2, whereas they did not have a significant impact on IgG-A7 binding to the Omicront. Furthermore, the location of P5E1-A6 and P5A10-G2 epitopes explained why the precursors of P5E1-A6 and P5A10-G2 (P5E1 and P5A10) had different binding profiles (see Table 2). Although a substantial overlap in P5E1-A6 and P5A10-G2 epitopes exists, P5E1-A6 has three contact clusters in the core of the RBD interface with hACE-2, whereas P5A10-G2 has only two. More as well as different contacts of P5E1-A6 and P5A10-G2 on the RBD should have led to the higher blocking activity and neutralization potency of P5E1-A6 with respect to P5A10-G2.

## 3. Discussion

In the previous sections, we reported the discovery and optimization of potent anti-SARS-CoV-2 neutralizing antibodies using ALTHEA Gold Plus Libraries™ as source of V regions, and the stepwise selection and optimization strategy summarized in Figure 5. In the first step, two lead antibodies, P5E1 and P5A10, were selected in a solid phase panning using RBD WT as a selector. These antibodies blocked the RBD:hACE-2 interaction and neutralized 80% of SARS-CoV-2 in vitro at 100 µg/mL. By reshuffling diversity at HCDR1 and HCDR2, in the second step, the affinity of the two antibodies was improved to the low-nanomolar range, which translated into 100% neutralization at 100 µg/mL and NC_50_ values in the low nanomolar range. In the third and final step, the affinity was improved further by reshuffling diversity at VL, leading to molecules with sub-nanomolar neutralization potency, slightly superior to IgG-A7, the potent anti-SARS-CoV-2 neutralizing antibody isolated from the immune library [8]. Importantly, the stepwise optimization process, which translated into superior neutralization potency, did not compromise the developability profile of the leading molecules, pointing to the high quality of the VH and VL library designs.

A comparison of the outcome of the selections from ALTHEA Gold Plus Libraries™ with the immune library reported elsewhere [8] offered a unique opportunity to assess the performance of a naïve, general-purpose library versus an immune library, with practical implications for antibody library design and high affinity antibody selection strategies. For instance, in this work, we isolated 24 positive and unique clones out of 630 assayed for RBD binding, for a hit rate of 3.8%. Panning of the immune libraries generated 34 unique out of 90 positive clones for RBD binding, for a hit rate of 38%. The difference of one order of magnitude in the immune library with respect to ALTHEA Gold Plus libraries™ was somewhat expected, as it reflects the bias in the immune repertoire towards RBD binders due to vaccination and SARS-CoV-2 Delta infection. In contrast, ALTHEA Gold Plus libraries™ are a non-biased representation of antigen-binding sites with the potential to generate specific antibodies to any given target. The price to pay is a relatively lower hit rate during the discovery phase.

By the same token, the antibodies obtained from the initial selections of ALTHEA Gold Plus Libraries™ resulted in relatively low affinity binders with suboptimal neutralization potency. Those obtained from the immune library were high affinity binders with neutralization potency comparable to EUA antibodies, e.g., CB6, selected from B-cell cloning [14]. However, the limitation of ALTHEA Gold Plus Libraries™ to generate highly potent antibodies in the discovery (initial) phase was mitigated by implementing a relatively simple process of sequentially reshuffling diversity at VH and VL, mimicking the natural affinity maturation process. With such a strategy, we were able to increase the affinity and neutralization potency of the antibodies to values slightly superior to those obtained from the immune library and with potency superior to CB6, the precursor of etesevimab, an EUA therapeutic antibody obtained from a B lymphocyte of a COVID-19 survivor.

The sequence analysis of the panels of antibodies selected from the immune library [8] and ALTHEA Gold Plus Libraries™ indicated that all the four VL scaffolds were seen in the unique clones. Thus, the design of the four VL libraries enabled the selection of diverse combinations of VLs with VH sequences specific for the RBD regardless of their origin, e.g., a universal naïve VH synthetic library or an immune VH repertoire evolved in vivo after vaccination and infection with SARS-CoV-2. This plasticity of VL to accommodate highly potent and diverse antigen-binding sites was consistent with its subsidiary role in determining the functional properties of the antibodies, with VH playing the leading role in binding a given epitope and thus determining the functionality of the antibodies. Interestingly, only a few mutations at VL were enough to improve the affinity of P5E1-A6 by two orders of magnitude, thus indicating that although VH determined the epitope and functionality of antibodies, diversification strategies focused on the CDRs of VL have a great potential for affinity improvement.

A detailed observation of the VL usage further revealed some commonalities and differences between this work and the selections performed from the immune library. Table 4 compares the use frequency of the VL scaffolds selected from ALTHEA Gold Plus libraries™ and the ones selected from the immune library. The 1-39 scaffold had roughly the same frequency (35–40%) in both selections. However, 1-39 was the predominant VL scaffold in the selections from ALTHEA Gold Plus libraries™, whereas, in the immune library, 3-20 outperformed 1-39 by almost 15% for a ~50% use frequency. Curiously, the use frequency of 3-20 and the other two scaffolds were more evenly distributed in the selections from ALTHEA Gold Plus libraries™ than in the immune library, where the frequency of 3-11 and 4-01 scaffolds was below 10%.

The 1-39 scaffold was the best one performing in the filtration process to generate ALTHEA Gold Plus libraries™ (see Figure 1). This might explain the higher frequency of this scaffold in both libraries. The high frequency of the 3-20 scaffold in the immune library may have partially been due to the high use of this IGKV gene in human immune responses, which have been found in over 30% of all IGKV/IGKJ rearrangements in some individuals [22].

Regarding VH, six IGHV germline genes (1-69, 1-24, 3-23, 3-53, 3-9 and 1-46) were found in the immune libraries. Since ALTHEA Gold Plus libraries™ were built with the universal IGHV3-23 germline gene, all the antibodies were variants of this gene. Notably, similar to VL, the HCDR3 of the antibodies isolated from ALTHEA Gold Plus libraries™ (Figure 6) were more evenly distributed than those of the immune libraries. No HCDR3 reached more than 20% use frequency, in contrast to the skewed HCDR3 lengths of the antibodies isolated from immune libraries where one length was seen in almost 50% of the antibodies. Moreover, the HCDR3 lengths of antibodies isolated from ALTHEA Gold Plus libraries™ were more diverse (13 lengths) than those selected from the immune libraries, with only seven lengths. Furthermore, the HCDR3 lengths in ALTHEA Gold Plus libraries™ were biased toward relatively short loops, with the most frequent HCDR3 being 9 residues, whereas the most prevalent HCDR3 lengths from the immune library were relatively long HCDR3 (15 residues). Therefore, a limited diversity at VH of ALTHEA Gold Plus libraries™ due to the use of a single universal scaffold seemed to have been compensated by a more diverse repertoire of relatively short HCDR3 loops. Of note, although long HCDR3 loops have been the landmark of neutralization in viral infections, such long loops tend to generate less stable antibodies and thus, the resultant antibodies are less developable than those with short HCDR3 loops.

Finally, one can argue that potent neutralizing antibodies, e.g., IgG-A7, were obtained from the immune library with a lesser effort, i.e., a fewer number of clones were screened and characterized, when compared to ALTHEA Gold Plus Libraries™. Therefore, the immune library seemed to be a more efficient means to discover V regions for diagnostic and/or antibody-based drug development. Nonetheless, it is worth mentioning that although we characterized the outcome of each step in the stepwise discovery and optimization strategy, VH and VL RAM are amenable to automation without the need of intermediate characterization nor design of new libraries—we used the PLs as source of diversity in the RAM approach, which should speed-up the optimization process of antibodies selected during the discovery phase. In comparison, making immune libraries requires: (1) the identification of a donor with high titers of specific antibodies, (2) the collection of B-cells, (3) the RT-PCR repertoire of V regions, (4) to synthesize the libraries in a scFv or Fab format by PCR or molecular cloning, (5) cloning and electroporation of the libraries, and (6) library rescue and quality control. In our experience, these tasks may take, depending on the laboratory expertise, 4–6 weeks, whereas naïve libraries are “ready-to-use” and, hence, bypass all the above steps.

Of note, potent neutralizing anti-SASR-CoV-2 antibodies from naïve libraries with similar or even higher potency than those obtained from immune libraries have recently been reported [6]. The authors performed two rounds of selection from a semisynthetic phage display naïve library first, followed by yeast display sorting. This strategy combined the advantages of large and diverse naïve phage display libraries with a precise subsequent yeast display selection using flow cytometry. Importantly, few developability liabilities were identified in the selected antibodies, emphasizing the value of well-designed naïve libraries to accelerate the discovery, manufacturing and clinical testing of antibodies isolated from this type of libraries.

In summary, we have demonstrated that naïve general-purpose synthetic VH repertoires produce similar results than those obtained from an immune VH repertoire. Since naïve libraries are “ready-to-use”, their use avoids the search for immune repertoires and library preparation, thus potentially speeding up the isolation of valuable antibodies in quickly spreading and evolving viruses. In addition, synthetic and semisynthetic naïve libraries can be designed with well-expressed and developable scaffolds, leading to antibodies amenable to a fast development process, thus reducing the timeframe between discovery and clinical testing. A shorter antibody development process is critical in the quest for diagnostic, prophylactic and therapeutic options to control not only SARS-CoV-2 but also other infection diseases that may emerge or resurface in the near future. 

## 4. Materials and Methods

### 4.1. Selection against the RBD WT of SARS-CoV-2

Expression and purification of RBD Wuhan (WT) have been described in detail in our previous publications [8,23]. Panning was performed as in Mendoza-Salazar et al. [8] with slight modifications. In brief, 8 wells of a Nunc Maxisorb plate were coated overnight with RBD at 50 µg/mL in PBS at 4 °C. The next day, the coated wells were washed three times with PBS and blocked with 3% skimmed milk prepared in PBS (MPBS 3%). The first round of panning was performed by incubating 100 µL per well of 1 × 10^12^ virions/mL (2 mL total) of each the four ALTHEA Gold Plus Libraries™ separately. For the subsequent rounds of selection, the concentration of RBD was reduced to 25 µg/mL. Rounds 2 and 3 were performed with mixed outputs of the previous rounds. To elute specific phages, 1 mg/mL of trypsin TPCK-treated (Sigma Aldrich, St. Louis, MO, USA, Cat. T1426) was added to wells and incubated for 10 min. An additional elution step was performed with Glycine-HCl pH 2.2 at room temperature (RT). Eluted phages from both phage elutions were mixed and amplified in with *E. coli* TG1. The amplified phages were rescued with Helper phage CM13K (ADL, Cat. PH050L).

### 4.2. Rapid Affinity Maturation (RAM)

RAM VH libraries consisted of combining the VL chain and HCDR3 of selected antibodies with VH synthetic fragments containing diversified HCDR1 and HCDR2 sequences before the filtration process. The resulting scFvs were cloned into a pADL23c vector and electroporated in *E. coli* TG1. The quality of the libraries was assessed by sequencing ten randomly chosen clones after electroporation. All clones had the proper scFv configuration and were different at the CDRs, and there were no mutations in the framework regions. The RAM VL library was built similar to RAM VH except that the VH chain of selected antibodies was amplified by PCR and assembled by overlapping PCR with synthetic VL fragments before the filtration process.

Different from the discovery campaign, affinity improvement using the RAM libraries was performed by solution panning. To this end, 100 µL of Streptavidin Magnetic Beads (Dynabeads™ M-280 Streptavidin, Thermo Fisher Scientific, Waltham, MA, USA) were incubated with each RAM-library to subtract non-specific phages. Biotinylated RBD WT (Acro Biosystems, Newark, DE, USA) was used as a selector in decreasing 10, 1 and 0.1 nM concentrations for rounds 1, 2 and 3, respectively. The first selection round was performed with 1 × 10^12^ virions in 3% PBS-BSA and blocking 1 h at RT. Then, biotinylated RBD was added to the phages and incubated for one additional hour at RT. The complex protein-phages were pulled down with streptavidin beads and washed with PBS-T (PBS with 0.1% Tween-20) and PBS. For rounds 2 and 3, 100 nM of non-biotinylated RBD were added to the beads and incubated overnight at 4 °C. After the washing, specific phages were eluted as described above.

### 4.3. Screening for Functional Clones

The screening for functional clones was performed as described in Mendoza-Salazar et al. [8]. In short, colonies were selected from 2YT plates of Round 3 and grown in 2 mL Nunc™ DeepWell plates (Thermo Scientific™, Cat. 278743) containing 2xYT with Glucose (1%) and Carbenicillin [100 µg/mL], and incubated overnight at 37 °C. On the next day, the scFvs were induced with IPTG 1 mM final concentration with an overnight incubation at 30 °C. The supernatants were tested for: (1) scFv expression using a Protein L/anti-myc HRP assay, (2) binding to RBD and (3) BSA (specificity control). Unique clones were determined by Sanger sequencing of the positive and specific clones and submitted for secondary screening.

### 4.4. Secondary Screening

The secondary screening consisted of: (1) blocking the RBD:hACE-2 interaction in an Intellicyt^®^ iQue3 system (Sartorius; Göttingen, Germany) as described by Mendoza-Salazar et al. [8] and (2) an alternative assay called competition assay, which was similar to the blocking assay except that the biotinylated RBD protein (SPD-C82E9, Acro Biosystems) was captured by the SAv (streptavidin) beads (iQue Qbeads^®^ DevScreen, Sartorius) and 20 µL of each antibody dilution plus 20 µL of 50 ng/mL biotinylated hACE-2 (AC2-H82E6, Acro Biosystems) were transferred to a 96 V-wells plate. Next, 10 µL of RBD-Qbeads were added. The Qbeads-RBD-hACE-2-biotin were detected with 10 µL of 1:500 dilution of Streptavidin-PE.

### 4.5. Expression, Conversion to hIgG1 Format, Purification and Control Antibodies

Cloning, expression and purification of the hIgG1 was performed as described by Mendoza-Salazar et al. [8]. As control antibodies, we used SARS-CoV-2 neutralizing antibody CB6 [14] as a positive control to set up the assays. CB6 is the precursor of Eli Lilly antibody Etesivimab [24]. The latter has the same V regions than CB6 and differ in the Fc region. In addition, we used IgG-A7 [8] as comparator antibody. The anti-lysozyme antibody D1.3 [25] was used as negative control.

### 4.6. Developability

Protein A purified antibodies were evaluated in an analytical platform to determine their potential for pharmaceutical development. This platform assessed identity/integrity by denaturing polyacrylamide electrophoresis (SDS-PAGE), purity by size exclusion chromatography (SEC) and thermal stability by Thermal Shift assay™. As previously described [13,23,26], all analytical techniques were performed using standard and well-known physicochemical methods for proteins.

### 4.7. Surface Plasmon Resonance (SPR)

Affinity was evaluated as described by Mendoza-Salazar et al. [8]. In short, the antibodies were captured on a Protein A CM5 sensor chip with the RBD WT flown over the immobilized antibodies. The association and dissociation rate constants were determined by fitting the raw data to a 1:1 Langmuir model using the BIAevaluation software (Version 3.1).

### 4.8. Plaque Reduction Neutralization Test (PRNT)

SARS-CoV-2 in vitro neutralization potency was assessed using a Plaque Reduction Neutralization Test (PRNT) as described by González-González et al. [9]. Briefly, Vero E6 cells (ATCC, Manassas, VA, USA, Cat. CRL-1586) were infected with SARS-CoV-2 obtained from clinical isolates (GenBank, accession number: OL790194; SARS-CoV-2/human/MEX/OAX P1/2020), in the presence of serial dilutions of the antibodies. After an incubation, lytic plaques were quantified, and the percentage of neutralization was calculated with respect to the infected and uninfected controls. This assay was performed in BSL2+ facilities, with strict biosafety standards and risk assessment protocols according to the specifications of the WHO Laboratory Biosafety Manual, Fourth Edition and the Guidance for General Laboratory Safety Practices during the COVID-19 Pandemic of CDC [27,28,29,30].

### 4.9. Epitope Mapping

The epitopes on RBD recognized by P5E1-A6, P5A10-G2 and IgG-A7 antibodies were mapped using CovalX epitope mapping service based on mass spectrometry cross-linking DSS MALDI MS analysis (https://covalx.com/crosslinking-epitope-mapping-service.php; accessed on 15 January 2023).

## Figures and Tables

**Figure 1 ijms-24-04609-f001:**
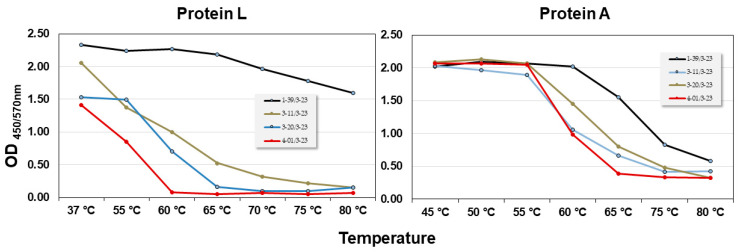
Comparison of the thermal stability profile of the VL scaffolds using Protein L and Protein A.

**Figure 2 ijms-24-04609-f002:**
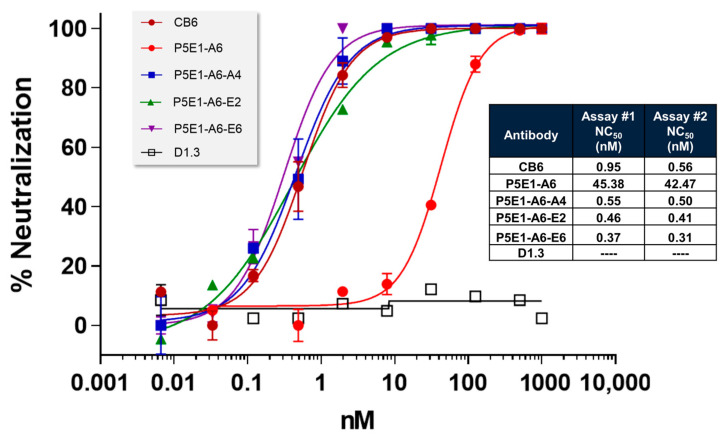
Authentic SARS-CoV-2 neutralization test of VH and VL optimized antibodies as compared to the parental antibody (P5E1-A6) and CB6. The table at the right of the plot reports the value of the neutralization potency of two assays performed in different days. The plot corresponds with Assay #1 dataset.

**Figure 3 ijms-24-04609-f003:**
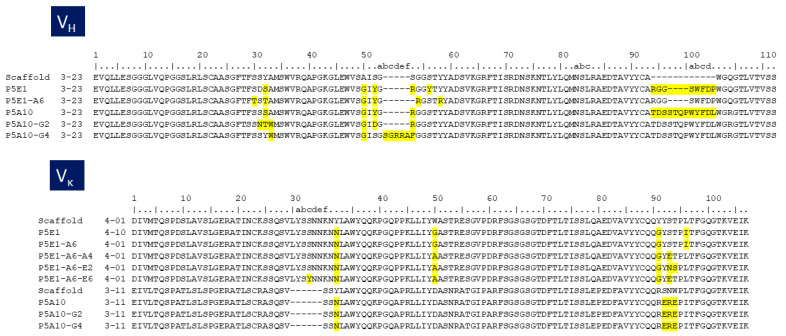
Alignment of the amino acid sequences of parental molecules (P5E1 and P5A10), VH RAM optimized (P5E1-A6, P5A10-G2 and P5A10-G4) and the final VL RAM optimized molecules (P5E1-A6-A4, P5E1-A6-E2 and P5E1-A6-E6). The second column corresponds to the VH or VL scaffolds. The mutations with respect to the VH scaffolds are highlighted in yellow. Numbering on top of the sequences followed the conventions of Chothia et al. [18].

**Figure 4 ijms-24-04609-f004:**
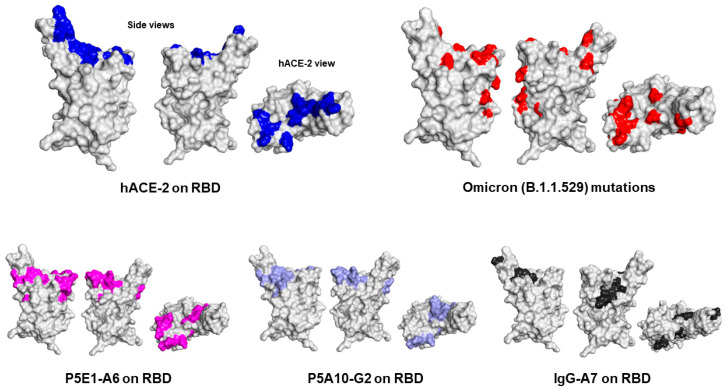
Epitope of P5E1-A6 (Violet), P5A10-G2 (light blue) and IgG-A7 (Black) on the Connolly surface of SARS-CoV-2 RBD WT. As a reference, we show, on the top left, two side views of the RBD rotated 180° and the top view of RBD interface (dark blue) with the hACE-2 (hACE-2 view). On the top right, Omicron mutations (red) with respect to Wuhan RBD. The figures were prepared with the PDB ID: 7SWP in Discovery Studio 2020 v20.1.0.19295 (BIOVIA).

**Figure 5 ijms-24-04609-f005:**
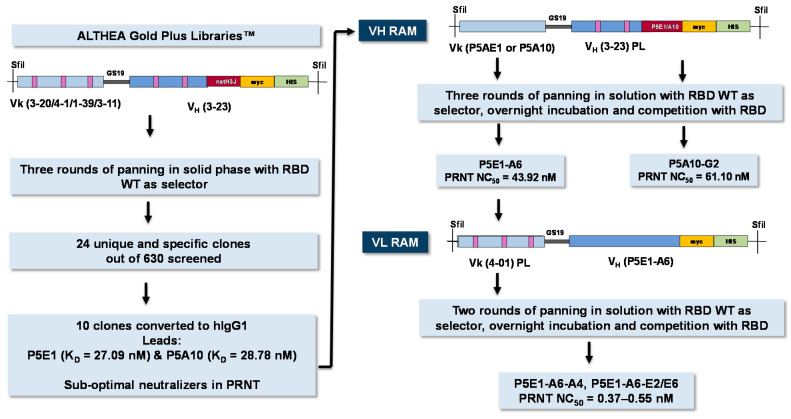
Stepwise strategy to isolate potent neutralizing antibodies from ALTHEA Gold Plus libraries™.

**Figure 6 ijms-24-04609-f006:**
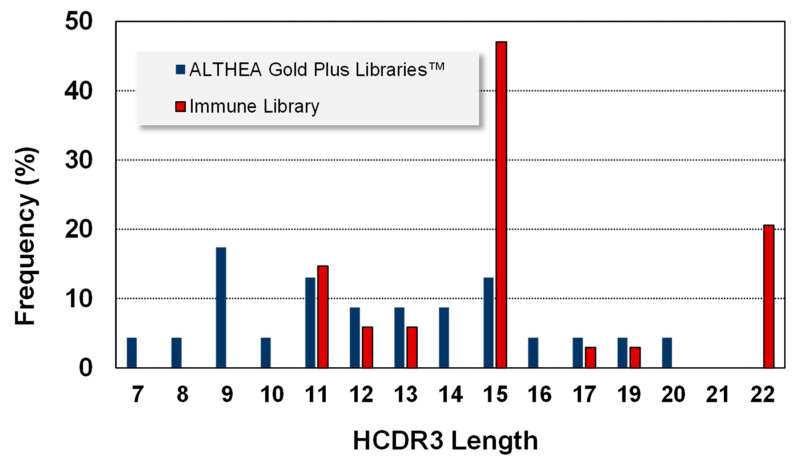
HCDR3 length frequency of the selected clones from ALTHEA Gold Plus libraries™ and the immune library [8].

**Table 1 ijms-24-04609-t001:** ALTHEA Gold Plus Libraries™ selection outcome after three rounds of panning with RBD WT.

Name	Frequency	VL Scaffold	HCDR3-Length (Kabat Definition)	HCDR3 Sequence	O.D. (450/570 nm)
E4R3	64	1-39	14	DGSSGWYKGGAFD	0.66
P4A9	15	1-39	11	VGGQWLDAFDI	0.87
F6R3	6	1-39	9	DRGNDAFDI	2.06
P1F2	5	1-39	12	VDYGDYGYSFDF	1.56
E3R3	4	3-11	9	GIHGEAFDY	0.57
G5R3	4	3-11	10	DRTAYGGNDY	0.25
P2F5	3	3-11	15	VYPYYYDSSGYVVDY	1.38
P2C8	3	3-20	20	DARSSSIAAWVHPDDYGMDV	0.40
D4R3	3	1-39	12	VDYGDYGYSFDY	0.18
P3G1	2	3-20	17	RDAIYGDYVPDYYGMDV	2.53
P3D1	2	ND*	19	WDCSGGSCYPSTYYYGMDV	1.93
D3R3	2	1-39	14	DGSSGGYKGGAFDI	0.42
B4R3	1	1-39	9	ENHWDAFDI	0.48
P5A10	1	3-11	11	DSSTQPWYFDL	3.35
P5E1	1	4-01	7	GGSWFDP	2.38
P1C8	1	1-39	9	VRHYYGMDV	1.61
P2E5	1	3-11	11	DLNVPAAIFGY	1.53
P2A3	1	ND	13	GTVGVQSGDAVDI	1.24
P3A9	1	ND	14	EGSSGWCKGGAFDI	0.68
A10R3	1	1-39	9	GIHGEAFDY	0.41
F7R3	1	4-01	16	EAYDYEGSGSEKAFDI	0.32
P5C6	1	4-01	8	DMGMGADY	0.17
C8R3	1	3-20	15	EKAGGNGWSYDAFDI	0.23
C4R3	1	ND	13	GTYDFWSGYSVDY	0.13

ND: Not determined.

**Table 2 ijms-24-04609-t002:** Summary of characterization of P5E1 and P5A10.

Assay		Units	P5E1	P5A10	IgG-A7 (1)	CB6
Monomeric Content		%/kDa	95.6/161	96.8/177	100/138.1	94.7/146.0
Thermal Stability	Tm1	°C	64.3	69.3	68.5	69.3
	Tm2	°C	83.6	82.0	82.1	82.0
ELISA	EC_50_	nM	0.97	16.87	0.025	0.027
	ka	1/Ms	1.60 × 10^5^	1.27 × 10^5^	5.20 × 10^5^	1.10 × 10^6^
SPR	kd	1/s	4.33 × 10^−3^	3.67 × 10^−3^	3.55 × 10^−4^	1.32 × 10^−2^
	K_D_	nM	27.09	28.78	0.68	12.06
Competition	IC_50_	nM	2.15	2.34	0.19	0.49
Blocking	IC_50_	nM	2.63	--	0.25	0.30–1.29
Neutralization	NC_50_	nM	80% neutralization at 100 µg/mL	80% neutralization at 100 µg/mL	0.56	0.56–2.74 (2)

(1) Data reported in [8,9]. (2) A range of values is reported to capture the variability of several neutralization assays reported here (see Figure 2) and those reported in a previous publication [8].

**Table 3 ijms-24-04609-t003:** Summary of characterization of P5E1-A6 and P5A10-G2 and P5A10-G4.

Assay		Units	P5E1-A6	P5A10-G2	P5A10-G4
Monomeric Content		%/kDa	100/151.0	100.0/165.0	99.1/151.0
Thermal Stability	Tm1	°C	68.2	68.5	64.3
	Tm2	°C	78.2	81.1	76.3
	ka	1/Ms	5.88 × 10^5^	1.49 × 10^6^	1.09 × 10^6^
SPR	kd	1/s	5.27 × 10^−4^	5.78 × 10^−4^	1.48 × 10^−4^
	K_D_	nM	0.89	0.39	0.14
Competition	IC_50_	nM	0.06	1.23	0.12
Blocking	IC_50_	nM	0.61	1.18	0.78
Neutralization	NC_50_	nM	43.92	61.10	10.45

**Table 4 ijms-24-04609-t004:** VL scaffold usage.

Scaffold	ALTHEA Gold Plus Libraries™	Immune Library
1-39	38%	35%
3-11	21%	6%
3-20	13%	47%
4-01	13%	9%

## Data Availability

This manuscript includes all the data used to conclude our research. Additional information is available on reasonable request to the corresponding author.

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
