# Peer review of "Discovery and Optimization of Neutralizing SARS-CoV-2 Antibodies Using ALTHEA Gold Plus Libraries™"

_ijms, 2023, doi:10.3390/ijms24054609_

Round 1

Reviewer 1 Report

Guzmán-Bringas et al, reported a unique antibody phage display selection method for isolating mAbs against SARS-2-CoV viruses. The utilized a stepwise selection and optimization strategy to select well expressed, high affinity and thermally stable mAb candidates with sub-nanomolar neutralization potency. The described technique can expedite the isolation of promising mAbs against highly mutated viruses.

However, there are few comments:

1-    In the sequence analysis section, the authors discussed P5E1-A6 and P5A10-G4, but not P5A10-G2. Could the authors explain more?

2-    Why the authors optimise the VL of P5A10-A6 using RAM, despite not binding to Omicron’s RBD (Line 279)?

3-    The following sentence if I’m not mistaken contradict itself for molecule P5E1-A6 “…P5E1-A6 binds the RBDs of Wuhan and Delta, but not Omicron. While P5E1-A6 and P5A10-G2 bind several of the residues mutated in Omicron.” Could the authors clarify this? Does this mean both P5E1-A6 and P5A10-G2 bind to Omicron?

4-    It would be useful and interesting to compare the report outcomes with Peng HP et al. 2014 report (doi: 10.1073/pnas.1401131111. Epub 2014 Jun 17. PMID: 24938786), in terms of CDRs/paratopes amino acids contents. Is it compatible with the Peng HP et study?

Author Response

Reviewer:   

Guzmán-Bringas et al, reported a unique antibody phage display selection method for isolating mAbs against SARS-2-CoV viruses. The utilized a stepwise selection and optimization strategy to select well expressed, high affinity and thermally stable mAb candidates with sub-nanomolar neutralization potency. The described technique can expedite the isolation of promising mAbs against highly mutated viruses.

However, there are few comments:

In the sequence analysis section, the authors discussed P5E1-A6 and P5A10-G4, but not P5A10-G2. Could the authors explain more?

Response:

Added - see lines 268-269 in the redlined manuscript: “P5A10-G2 had three mutations in the HCDR1 and one in HCDR2 with respect to the parental P5A10 but did not have indels”.

Reviewer:   

Why the authors optimise the VL of P5A10-A6 using RAM, despite not binding to Omicron’s RBD (Line 279)?

Response:

Please notice that the main goal of this work is to compare the selections of an immune library versus a naïve library. Since we used RBD Wuhan as selector in the naïve library we did the same with the immune library. Therefore, we focused on the results obtained with SARS-CoV-2 Wuhan and optimization of the antibodies regardless its cross-reactivity with Omicron.

Reviewer:   

The following sentence if I’m not mistaken contradict itself for molecule P5E1-A6 “…P5E1-A6 binds the RBDs of Wuhan and Delta, but not Omicron. While P5E1-A6 and P5A10-G2 bind several of the residues mutated in Omicron.” Could the authors clarify this? Does this mean both P5E1-A6 and P5A10-G2 bind to Omicron?

Response:

It is confusing indeed.  It is rephrased in the revised version of the manuscript as:Elucidation of the epitopes recognized by P5E1-A6, P5A10-G2 and IgG-A7 also shed light on previously reported functional data [8] showing that: (1) P5E1-A6 competes with IgG-A7 for binding to RBD and (2) IgG-A7 binds RBDs of Wuhan, Delta and Omicron, whereas P5E1-A6 binds the RBDs of Wuhan and Delta, but not Omicron. P5E1-A6 and P5A10-G2 epitopes on the RBD show that these antibodies bind several of the residues in the RBD WT that are mutated in Omicron. In contrast IgG-A7 epitope mostly maps onto residues in RBD WT that have not been changed in Omicron. Therefore, Omicron mutations seemed to have abrogated binding to P5E1-A6 and P5A10-G2, whereas have not had a significant impact on IgG-A7 binding to Omicron.”

Reviewer:   

It would be useful and interesting to compare the report outcomes with Peng HP et al. 2014 report (doi: 10.1073/pnas.1401131111. Epub 2014 Jun 17. PMID: 24938786), in terms of CDRs/paratopes amino acids contents. Is it compatible with the Peng HP et study?

Response: 

The paper suggested by the reviewer studies the functional paratope predictions on a set of 111 antibody–antigen complex structures and found that aromatic, mostly tyrosyl, side chains constitute the major parts of the predicted functional paratopes, with short-chain hydrophilic residues forming the minor portion of the predicted functional paratopes.  

Consistent with the above paper, we found several tyrosine residues and short side hydrophilic residues such as Serine, threonine and asparagine in the CDRs of the sequences shown in Figure 3 of our manuscript. However, we did not discuss this finding since the library design took in consideration the literature on amino acid content in the CDRs - it is discussed in detail in the paper where we described the design of ALTHEA Gold  libraries – see: Valadon//Almagro. MABS 2019, VOL. 11, NO. 3, 516–531. https://doi.org/10.1080/19420862.2019.1571879.

Quoting from the discussion of Valadon//Almagro’s paper:

“ … analyses of the ALTHEA Gold Libraries™ construction process by NGS revealed important lessons for future library designs. Over a million sequences obtained by NGS from the PLs and FLs indicated an increase in polar residues and a decrease in hydrophobic and Y residues in some designed positions after submitting the PLs for heating at 70°C for 10 min and rescuing the well-folded variants with Protein A. Mapping these positions in the structure of the scaffolds indicated that they are spatially close, and thus the changes observed upon filtration may be correlated. In a previous work, [19] we noticed that polar amino acids, particularly R, N, D, S, T and G frequently occur at the antigen-binding site. Non-polar amino acids such as A, I, V, M, L and F are significantly underrepresented. Y was found in a high proportion in the antigen-binding site, consistent with previous studies. For instance, Lo Conte et al. [40] observed that Y contributed to 17% of all amino acids in contact in the 19 antigen-antibody complexes available at that time. Similarly, an earlier work by Mian et al. [41] reported the overuse of Y to contact antigens. Based in part on these early works and the observation that Y is a versatile amino acid, capable of generating diverse interactions with the antigens, Sidhu et al. [42] elegantly showed that by creating antibody libraries with only Y and S or W and S, specific antibodies with affinity in the nM range can be obtained, similar to those obtained from more complex libraries.”

Reviewer 2 Report

General comments

This article describes an experimental method for finding neutralizing antibodies targeting the receptor binding domain (RBD) of SARS-CoV-2 using large pools of designed antibody sequence libraries. A version of this approach was previously developed to target SARS-CoV-2. This work implements the concept of a general sequence library designed without the knowledge of immune libraries derived from infected individuals. The approach was tested against Wuhan, wild-type RBD to produce a couple of antibodies having about the same potency as a known neutralizing antibody (CB6). While the general library approach of the work may have interesting applications, the efficacy of the method has not been tested against circulating SARS-CoV-2 variants, as well as other targets. 

Specific comments

1. The general library approach requires additional layers of screening/selection (RAM) to improve RBD affinity and neutralization potency. The RAM steps also require construction of focused libraries. The generality of these protocols is unclear. For example, will the protocols work for other targets (other than spike RBD)? The authors should critically discuss the advantages and disadvantages of their approach versus the use of immune libraries in terms of resources needed, simplicity of implementation, and speed.

2. In Fig. 2, the NC50 (concentration at 50% neutralization) values for ASSAY #2 in the table and curves do not seem to agree for antibodies P5E1-A6-E2/6.

3. In Fig. 3, should the competition be with hACE2 rather than the RBD?

4. In Fig. 4, it is not clear how the RBD recognition sites for the discovered antibodies are determined. Are they based on previous or predicted recognition sites?

Minor comments

Line 19: change "One the..." to "One of the..." 

Line 68: Ref 5 should be ref 9? Also, Ref 9 should be Antibodies 2022, 11(57), and Ref 11, Antibodies 2022, 11(13)

Line 263: "...since compited with P5E1-A6...", not clear what this means.

Ref 10 is not accessible based on the information provided.

Author Response

Reviewer:

General comments:

This article describes an experimental method for finding neutralizing antibodies targeting the receptor binding domain (RBD) of SARS-CoV-2 using large pools of designed antibody sequence libraries. A version of this approach was previously developed to target SARS-CoV-2. This work implements the concept of a general sequence library designed without the knowledge of immune libraries derived from infected individuals. The approach was tested against Wuhan, wild-type RBD to produce a couple of antibodies having about the same potency as a known neutralizing antibody (CB6). While the general library approach of the work may have interesting applications, the efficacy of the method has not been tested against circulating SARS-CoV-2 variants, as well as other targets. 

Response: 

This is valid point and we have tested this approach in the discovery and optimization of several unrelated targets including TNFa and DP-1, with similar results to the ones described in the manuscript.  Since the scope of the paper was to compare the selection of an immune versus naïve libraries, in our opinion the validation of the approach with other targets is beyond the scope of the paper.

Reviewer:

Specific comments

The general library approach requires additional layers of screening/selection (RAM) to improve RBD affinity and neutralization potency. The RAM steps also require construction of focused libraries. The generality of these protocols is unclear. For example, will the protocols work for other targets (other than spike RBD)? The authors should critically discuss the advantages and disadvantages of their approach versus the use of immune libraries in terms of resources needed, simplicity of implementation, and speed.

Response:

We see two points in this comment:

  1. “RAM steps also require construction of focused libraries.”

RAM involves a cloning step, but it does not include a new focused library design and synthesis, which take several weeks. We do use the primary libraries (PLs) as source of diversity in RAM. The PLs are ready to use and importantly, have not been biased by the heat shock nor cloning with the VH/VL and natural H3J fragments which erode the designed diversity of 106 variants of the PLs (see Valadon//Almagro. MABS 2019, VOL. 11, NO. 3, 516–531. https://doi.org/10.1080/19420862.2019.1571879). Thus, by reshuffling existing and ready to use diversity, and showing that we can improve affinity and neutralization potency, the manuscript provided a fast method that avoided redesigning and synthesizing libraries, as is commonly practiced in traditional affinity maturation strategies.

  1. “critically discuss the advantages and disadvantages of their approach versus the use of immune libraries in terms of resources needed, simplicity of implementation, and speed”

We modified the discussion in the revised version of the manuscript to include a comparison of resources, simplicity of implementation, and speed of our approach vs immune libraries, as suggested by the reviewer.

Quoting from the revised version:

“ … one can argue that potent neutralizing antibodies, e.g., IgG-A7, were obtained from the immune library with a lesser effort, i.e., a fewer number of clones had to be screened and characterized, when compared to ALTHEA Gold Plus Libraries™. Therefore, the immune library seemed to be a more efficient means to discover V regions for diagnostic and/or antibody-based drug development. Nonetheless, worth mentioning is that although we characterized the outcome of each step in the stepwise discovery and optimization strategy, VH and VL RAM are amenable to automation without the need of intermediate characterization nor design of new libraries – notice that we used the PLs as source of diversity in the RAM approach, which should speed-up the optimization process of antibodies selected during the discovery phase. In comparison, making immune libraries require: (1) identification of a donor with high titers of specific antibodies, (2) collect B-cells, (3) RT-PCR the repertoire of V regions, (3) synthesize the libraries in a scFv or Fab format by PCR or molecular cloning, (4) cloning and electroporation of the libraries, and (5) library rescue and quality control. In our experience, these tasks may take, depending on the laboratory expertise, 4-6 weeks whereas, naïve libraries are “ready-to-use” and hence, bypass all the above steps.”

Also,

“Of note, potent neutralizing anti-SASR-CoV-2 antibodies from naïve libraries with similar or even higher potency than those obtained from immune libraries have recently been reported [6]. The authors performed two rounds of selection from a semisynthetic phage display naïve library first, followed by yeast display sorting. This strategy combined the advantages of large and diverse naïve phage display libraries with a precise subsequent yeast display selection using flow cytometry. Importantly, few developability liabilities were identified in the selected antibodies, emphasizing the value of well-designed naïve libraries to accelerate the discovery, manufacturing, and clinical testing of antibodies isolated from naïve libraries.”

Reviewer:

  1. In Fig. 2, the NC50 (concentration at 50% neutralization) values for ASSAY #2 in the table and curves do not seem to agree for antibodies P5E1-A6-E2/6.

Response:

We truly appreciate this comment as the difference between the reported EC50 values in the table and a visual inspection of Figure 2 may certainly lead to confusion. The values in the table were calculated with Prisma (https://www.graphpad.com/support/faq/prism-3-analyzing-ria-and-elisa-data/using) using a 4-paramenter mathematical model that takes into account the range of concentrations, OD signals, baseline, and saturation value to calculate the EC50.

The concentration range is the same for all antibodies. OD signals, baseline, and saturation value change from antibody to antibody. In our case, all the samples reached a similar saturation value but in Assay #2 not all the samples had the same baseline. Thus, the calculated EC50 vs the plotted EC50 slightly differ. 

The calculated EC50s vs the plotted EC50s in assay #1 are more consistent but we selected assay#2 to present in the manuscript as it better resolved the differences between antibodies.

In view of the referee’s comment, we have replaced assays #2 with assays #1 in the revised version of the manuscript. Notice that the baseline is similar in all the antibodies and the calculated EC50 values in the table and those plotted are very close.

Reviewer:

  1. In Fig. 3, should the competition be with hACE2 rather than the RBD?

Response:

We are not following well this comment as Figure 3 is the alignment of sequences.

Reviewer:

  1. In Fig. 4, it is not clear how the RBD recognition sites for the discovered antibodies are determined. Are they based on previous or predicted recognition sites?

Response:

The RBD recognition sites for the discovered antibodies were experimentally determined using CovalX’s epitope mapping service based on mass spectrometry cross-linking DSS MALDI MS analysis. At the end of the Material and Methods section, under subsection “4.7. Epitope mapping” we mentioned the method used to determine the epitopes. However, following the reviewer’s comment, we added more information in the revised version of the manuscript – see lines 604-609 of the redlined version of the manuscript.

Minor comments

Reviewer:

Line 19: change "One the..." to "One of the..." 

Response:

Corrected.

Reviewer:

Line 68: Ref 5 should be ref 9? Also, Ref 9 should be Antibodies 2022, 11(57), and Ref 11, Antibodies 2022, 11(13)

Response:

Revised and corrected all the references as needed.

Reviewer:

Line 263: "...since compited with P5E1-A6...", not clear what this means.

Should read “competed” – corrected in the manuscript.

Reviewer:

Ref 10 is not accessible based on the information provided.

Response:

Revised and corrected all the references as needed.

Reviewer 3 Report

In the manuscript "Discovery and optimization of neutralizing SARS-CoV-2 antibodies using ALTHEA Gold Plus Libraries" the authors demonstrate that  the ALTHEA Gold Plus Libraries can be a useful tool to develop antibody with a distinct antiviral effect against SARS-CoV-2. The manuscript is well written and the experimental design is fine. However, I have some comments that need further attention before the manuscript can be accepted for publication in IJMS.

1. Could the authors also add further information of the used SARS-CoV-2 strain.

2. Please perform additional experiments with the different VOCs to validate the antiviral effect of the used antibody. Are strain dependencies observed?

minor point

1. Please check the formatting as the figure legend of fig 6 is not in line with the previous ones.

The manuscript is well written and the data are interesting for a broad audience. I hope the authors can address all my concerns and revise the manuscript accordingly.

Author Response

Reviewer:

In the manuscript "Discovery and optimization of neutralizing SARS-CoV-2 antibodies using ALTHEA Gold Plus Libraries" the authors demonstrate that the ALTHEA Gold Plus Libraries can be a useful tool to develop antibody with a distinct antiviral effect against SARS-CoV-2. The manuscript is well written and the experimental design is fine. However, I have some comments that need further attention before the manuscript can be accepted for publication in IJMS.

  1. Could the authors also add further information of the used SARS-CoV-2 strain.

The SARS-CoV2 strain we used is described in lines 594-595 of the redlined version of the manuscript. Notice that we added more information in the revised version to help find the whole genome sequence of the SARS-CoV2 strain we used.

  1. Please perform additional experiments with the different VOCs to validate the antiviral effect of the used antibody. Are strain dependencies observed?

We have tested the antibodies against SARS-CoV-2 Wuhan and Delta. We only presented and discussed Wuhan since the main goal of the work was to compare selections performed with the same target (RBD from SARS-CoV-2 Wuhan) in the immune library versus naïve libraries. Therefore, in our opinion, testing, presenting, and discussing results of neutralization tests with other SARS-CoV-2 strains is beyond the scope of the paper.

Reviewer:

minor point

  1. Please check the formatting as the figure legend of fig 6 is not in line with the previous ones.

Corrected the format of Figure 6 legend to be the same as Figure 1 and 2.

Round 2

Reviewer 2 Report

I have no further comments.

Reviewer 3 Report

The main focus was indeed to compare selections performed with the same target (RBD from SARS-CoV-2 Wuhan) in the immune library versus naïve libraries. However, as the authors have already perform the experiments with the delta strain they should further validate the rigidity of the method by additional using the delta strain.